# Time-resolved spectroscopic and electrophysiological data reveal insights in the gating mechanism of anion channelrhodopsin

Max-Aylmer Dreier [1,2], Philipp Althoff[1,2], Mohamad Javad Norahan[1,2], Stefan Alexander Tennigkeit[1,2], Samir F. El-Mashtoly [1,2], Mathias Lübben[1,2], Carsten Kötting [1,2], Till Rudack [1,2✉] & Klaus Gerwert [1,2✉]

Channelrhodopsins are widely used in optogenetic applications. High photocurrents and low current inactivation levels are desirable. Two parallel photocycles evoked by different retinal conformations cause cation-conducting channelrhodopsin-2 (CrChR2) inactivation: one with efficient conductivity; one with low conductivity. Given the longer half-life of the low conducting photocycle intermediates, which accumulate under continuous illumination, resulting in a largely reduced photocurrent. Here, we demonstrate that for channelrhodopsin-1 of the cryptophyte Guillardia theta (GtACR1), the highly conducting C = N-anti-photocycle was the sole operating cycle using time-resolved step-scan FTIR spectroscopy. The correlation between our spectroscopic measurements and previously reported electrophysiological data provides insights into molecular gating mechanisms and their role in the characteristic high photocurrents. The mechanistic importance of the central constriction site amino acid Glu-68 is also shown. We propose that canceling out the poorly conducting photocycle avoids the inactivation observed in CrChR2, and anticipate that this discovery will advance the development of optimized optogenetic tools.

[1] Biospectroscopy, Center for Protein Diagnostics (PRODI), Ruhr University Bochum, Bochum, Germany. [2] Department of Biophysics, Ruhr University Bochum, Bochum, Germany. ✉email: till.rudack@rub.de; klaus.gerwert@rub.de

Optogenetics is a relatively new, rapidly growing research field that implements genetically modified eukaryotic cells to elicit physiological effects triggered by visible or ultraviolet light[1,2]. Recent advances demonstrate the enormous potential for various medical applications, including treating or even curing blindness[3]. The initial discovery of channelrhodopsins (ChRs) in *Chlamydomonas reinhardtii*, as well as their identification as light-activated ion channels, represent the inception of optogenetics[4–9]. Since then, numerous natural and engineered relatives of these channels have been used as photobiological instruments to control voltage-activatable cells such as neurons[10]. In addition to the light-activatable channels, various ion pumps comprising the microbial rhodopsin superfamily[1], as well as caged effector proteins that use light–oxygen–voltage domains, or dimerizing systems that employ phytochromes or cryptochromes to stimulate protein–protein interactions[11], are utilized as optogenetic tools.

Microbial rhodopsins are versatile and act either as ion or proton pumps, signal sensors, or channel proteins. They all contain seven transmembrane helices and the chromophore retinal, which is deeply inserted into the protein membrane-embedded part. The retinal co-factor is covalently bound to a lysine residue to form a protonated Schiff base (SB). Photon absorbance by retinal leads to isomerization, which initiates a cyclic cascade of conformational and protonation changes termed as a photocycle, described by distinct, consecutive series of intermediate states[12]. These states are usually denoted J–O, following the reference of the well-studied proton pump, bacteriorhodopsin (bR)[13], which can be regarded as a microbial rhodopsin archetype.

In the case of optogenetically highly relevant ChRs, light absorption induces the transient formation of a conducting ion pore, thus initiating ion flow according to preexisting ion concentration gradients at the membranes of living cells[14–17]. ChRs are subdivided into cation-conducting ChRs (CCRs) and anion-conducting ChRs (ACRs)[18]. While CCRs and ACRs have the same membrane topology and a generally similar structure, they differ in their physiological effects. Usually, CCRs produce membrane depolarization (generation of action potentials) to activate cells, whereas ACRs hyperpolarize membranes (suppression of action potentials) to deactivate cells[19]. To perform these tasks, ideal optogenetic channels would comprise features such as (i) high sensitivity to activating light, (ii) fast channel gating (i.e., fast opening and closing), and (iii) intense photocurrents. However, most ChRs diverge from these ideal channel characteristics.

Recently, we studied the physiological and biophysical mechanisms of the well-investigated cation translocating channelrhodopsin-2 found in *C. reinhardtii* (*Cr*ChR2)[6,7,20–22]. After light activation, the photocurrent of *Cr*ChR2 exhibits an initial sharp peak that rapidly decreases to a much lower but stable level[23] (Supplementary Fig 1). This inactivation effect is attributed to the branched photocycle of *Cr*ChR2[15], as illustrated in Fig. 1a. In our previous work, this photocycle model of *Cr*ChR2 was obtained using a combination of spectroscopic and electrophysiological data, unifying earlier published models and eliminating former discrepancies[15]. We have demonstrated that upon light activation, the dark-adapted ground state *Cr*ChR2 either undergoes a photocycle with an efficiently conducting open state (Fig. 1a, left) or enters another light-adapted ground state, which leads to an alternative photocycle characterized by a less efficient ion conduction (Fig. 1a, right). The well-conducting cycle is characterized by an all-*trans*,C = N-*anti* → 13-*cis*,C = N-*anti*-isomerization of retinal, and is named dark-adapted *anti*-cycle. The formation of the light-adapted ground state is characterized by the photoisomerization of the all-*trans*,C = N-*anti* state to

form a novel 13-*cis*,C = N-*syn* retinal ground state. The elicited photocycle is therefore named light-adapted *syn*-cycle. The *syn*-cycle comprises a poorly conducting open state that slowly decays back to either the dark-adapted or light-adapted ground state and accumulates under continuous illumination, thus causing a substantial decrease in the photocurrent of *Cr*ChR2 (Supplementary Fig. 1). The subject of our current investigation is the first discovered natural ACR anion channelrhodopsin-1 (*Gt*ACR1) of the cryptophyte *Guillardia theta*[18]. At first sight, the structural and molecular similarities between *Cr*ChR2 and *Gt*ACR1 suggest related photocycle conformational changes. However, the amplitudes, opening and closing kinetics, and shapes of photocurrents generated during continuous illumination by *Cr*ChR2 and *Gt*ACR1 differ significantly (Supplementary Fig. 1)[24]. In contrast to the pronounced inactivation observed in *Cr*ChR2, *Gt*ACR1 exhibits a slight linear decrease in photocurrent (Supplementary Fig. 1). Moreover, the conducting states in the photocycles of *Cr*ChR2[15] (Fig. 1a, left) and *Gt*ACR1[24] (Fig. 1b, left) are distinct: based on electrophysiological and UV/VIS spectroscopic data (see Fig. 1a, left)[24], *Cr*ChR2 is conducting cations in the M-like (named $P_{390}^M$) and N-like (named $P_{520}^N$) intermediate states, while *Gt*ACR1 begins anion conduction during the transition from K to L state. Interestingly, in *Gt*ACR1, two transitions have been electrophysiologically identified to reflect the conducting state formation (indicated by the two half-lives for K → L in Fig. 1b, left). However, none of these transitions could be resolved in the reported UV/VIS data[24], indicating a complex behavior. The decay of L to M makes *Gt*ACR1 non-conducting. Subsequently, the M intermediate state decays to a N/O intermediate state, during which a low conductance is observed due to the reversible reaction between L and M. Finally, *Gt*ACR1 returns from N/O to its ground state[24]. Although *Gt*ACR1 is structurally more similar to *Cr*ChR2 than to bR, this modeled reaction series resembles the non-branched photocycle of bR. Thus, certain aspects regarding the detailed molecular gating mechanism of *Gt*ACR1 remain unclear. Despite their overall structural similarity, *Cr*ChR2[25] and *Gt*ACR1[26,27] exhibit several marked differences. For instance, *Gt*ACR1 contains a preexisting tunnel[27,28] that is not present in *Cr*ChR2[25]. In addition to a low sequence identity (24%), the crucial residues in *Cr*ChR2 that are functionally responsible for the inner gate (Glu-82, Glu-83, His-134, and Arg-268)[25], the SB counter ion (Glu-123)[29,30], and a prominent key residue for the *syn*-cycle (Lys-93)[15], are absent from *Gt*ACR1. In contrast, a critical player in the *Cr*ChR2 photocycle, Glu-90[15–17], is conserved in *Gt*ACR1 (Glu-68) and other known ACRs[31], although absent in bR. Moreover, within *Gt*ACR1, Glu-68 together with Asp-234 and Cys-102, may participate in channel gating as their mutations influence channel kinetics[24,32]; interestingly C102A evokes bistability of *Gt*ACR1[33].

Molecular insights into the *Gt*ACR1 photocycle are needed to understand the correspondences and distinctions between *Cr*ChR2 and *Gt*ACR1. Previous static infrared spectroscopic investigations of *Gt*ACR1 provided substantial insights into the K and L intermediates by trapping them at cryogenic temperatures[26,34]. However, a precise picture of the molecular processes illustrating dynamic information at a high temporal resolution, as it can be achieved with time-resolved Fourier-transform infrared (FTIR) spectroscopy[35,36], is lacking. Thus, in this study, we investigated the *Gt*ACR1 photocycle using step-scan FTIR spectroscopy at ambient temperature to identify the early photocycle states and elucidate the conducting pore formation at sub-microsecond resolution. Further, we explored whether *Gt*ACR1 has a light-adapted *syn*-cycle and its potential role in inactivation. A comprehensive understanding of the molecular mechanism of *Gt*ACR1 compared to *Cr*ChR2 will enrich our understanding of the channel gating of ChRs and the

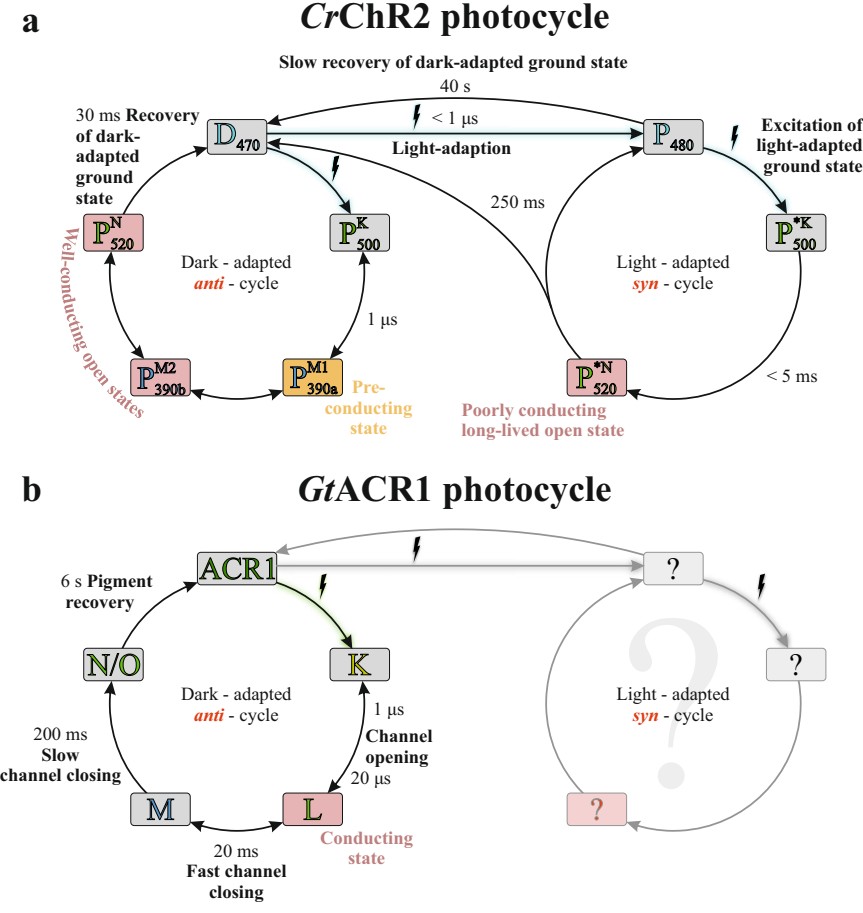

**Fig. 1 Photocycle models of *Cr*ChR2 and *Gt*ACR1. a** Model of the branched photocycle of *Cr*ChR2 (reproduced and modified from Kuhne et al.[15]). The "dark-adapted" cycle (left) exclusively comprises a C=N-*anti*-conformation of the retinal and two well-conducting open states that decay relatively fast. In the "light-adapted" ground state (right), the retinal has a 13-*cis*,C=N-*syn* conformation, and its photoproducts comprise a poorly conducting open state that decays at a slower rate. During continuous illumination, molecules are accumulated in the *syn*-cycle, which causes the inactivation of *Cr*ChR2. **b** Current photocycle model of *Gt*ACR1 (reproduced and modified from Sineshchekov et al.[24]), showing the ʟ-intermediate state as the open state followed by M and N/O state. The question of the existence of a branched photocycle as in *Cr*ChR2 is a topic of this investigation.

inactivation phenomenon. This understanding will advance the community in the design of improved optogenetic instruments exhibiting high-efficiency conductance while lacking the disadvantages associated with the inactivation effect.

## Results and discussion
To gain dynamic molecular insights into the photocycle of *Gt*ACR1, we performed a time-resolved spectroscopic investigation at ambient temperature. First, the photocycle of *Gt*ACR1 was studied by focusing on the early transitions related to gate opening by UV/VIS and vibrational spectroscopy. Second, the potential presence of a light-adapted state was excluded by means of a spectroscopic investigation of isotopically labeled retinal. From these data, we derived a revised photocycle model of *Gt*ACR1.

**Monitoring channel gating by time-resolved spectroscopy.** The time course of the photocyclic reaction series of *Gt*ACR1 was monitored over six orders of magnitude at ambient temperature by time-resolved UV/VIS and FTIR spectroscopy. Global fit analysis of the FTIR (Fig. 2a and Supplementary Figs. 2 and 3) and UV/VIS (Supplementary Fig. 4) data yielded half-lives for six partial reactions, which were named $T_1$–$T_6$. The half-lives and assignment of the reactions are summarized in Table 1, and a detailed description of the six processes is provided in

Supplementary Note 1. Overall, the first transition ($T_1$) represented the decay of the K-intermediate state of the photocycle. The final transition ($T_6$) represented a N/O mixed-species decay, back to the ground state. In contrast to the previous spectroscopic data (FTIR, UV/VIS), our determined half-lives of the other four transitions ($T_2$–$T_5$) closely matched those obtained from previous electrophysiological measurements[32]. This correlation enabled us to interpret the spectroscopically detected transitions $T_2$ through $T_5$ as gating processes. The amplitudes of the marker bands representing channel opening ($T_2$ and $T_3$) are larger in a faster rate, which agrees with the electrophysiological data that showed the first-rate of channel opening to be more prominent than the second rate[32]. Therefore, mechanistic molecular details such as conformational changes of the retinal, protonation state changes, as well as backbone and side-chain rearrangements, which are encoded in the spectroscopic data, were decoded in relation to the gating mechanism. Furthermore, we identified absorbance band differences, which correspond to the observed transitions, as marker bands for the respective processes.

**Marker bands for channel gating.** High spatiotemporal resolution methods to identify the characteristic bands for distinct events represent a key step toward understanding the molecular mechanisms that underlie channel gating. Time-resolved changes in absorbance bands correspond to the specific mechanistic

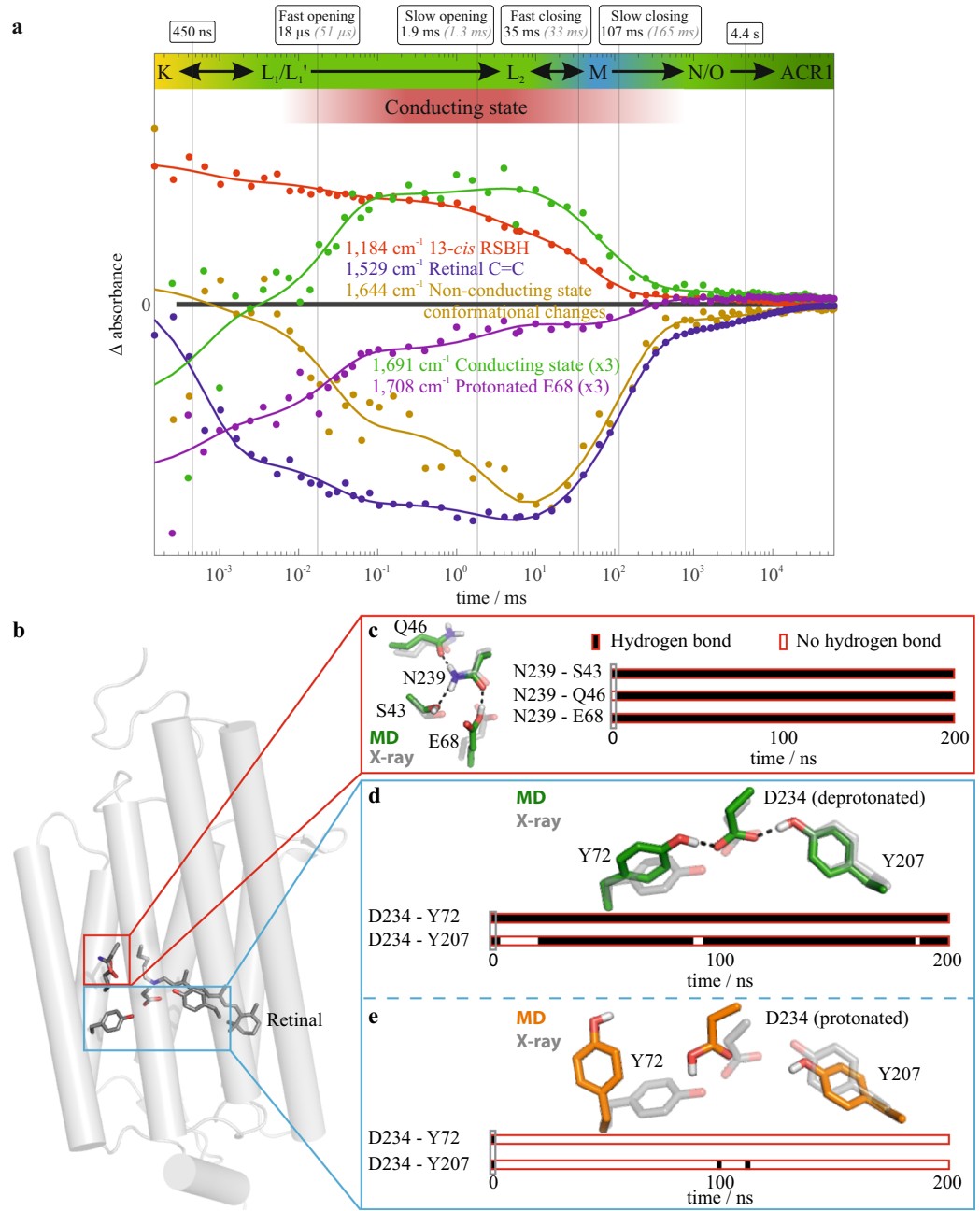

**Fig. 2 Time courses of marker absorption bands of *Gt*ACR1. a** Merged time-resolved data obtained from step-scan and rapid-scan FTIR difference spectroscopic measurements. The conducting state assignment is based on the half-lives obtained in the spectroscopic global fit analysis (bold black numbers and vertical solid black lines) and the published electrophysiologically determined half-lives of channel opening and closing[32] (gray italic numbers). The absorbance key bands were derived from the amplitude spectra (Supplementary Fig. 2). The 1644 cm$^{-1}$ band (ocher) reflects conformational changes during channel opening and closing in the amide I region. The 1691 cm$^{-1}$ band (green) corresponds to channel opening and serves as a marker band for the conducting state. The 1184 cm$^{-1}$ band (red) reflects the protonated 13-*cis* retinal previously established in microbial rhodopsins[37]. The 1529 cm$^{-1}$ band (blue) reflects the retinal C=C vibrations in the ground state[39]. Further kinetics are shown in Supplementary Fig 3. Based on these IR spectroscopic data along with UV/VIS data (Supplementary Fig. 3), we derived an extended *Gt*ACR1 photocycle model with a non-conducting L$_1$ and an opened L$_2$ state shown in Fig. 4. For a detailed discussion of all kinetic information, please refer to Supplementary Figs. 2–4 and 6 and Supplementary Note 1. **b** Overview of the *Gt*ACR1 ground state structure based on monomer A of protein data bank (PDB)-ID 6CSM[26]. **c** Representative hydrogen bond network of the central constriction site for protonated Glu-68 obtained using a molecular dynamics (MD) simulation initiated by monomer A of the X-ray structure 6CSM. **d** Representative structure (green) and contact analysis of a 100 ns trajectory initiated by monomer A of the X-ray structure 6CSM with deprotonated Asp-234 and Glu-68. The protonated SB is compared to the X-ray structure (gray). **e** Representative structure (orange) and contact analysis of a 100 ns MD simulation trajectory as described in (**d**) but with protonated Asp-234. The source data for **a**, **c**–**e** is given in Supplementary Data 1.

**Table 1 GtACR1 photocycle transitions.**

| Transition name | $t_{1/2}$ FTIR | $t_{1/2}$ UV/VIS | $t_{1/2}$ ephys[32] | $t_{1/2}$ UV/VIS[24] | Transition/process |
|---|---|---|---|---|---|
| $T_1$ | 450 ns | 3 µs | n.r. | 3.5 µs | K → L |
| $T_2$ | 18 µs | 23 µs | 51 µs | n.r. | Within L fast channel opening observed |
| $T_3$ | 1.9 ms | 2.3 ms | 1.3 ms | n.r. | Within L slow channel opening observed |
| $T_4$ | 35 ms | 23 ms | 33 ms | 15 ms | $L_2$ → M channel closing, fast channel closing observed |
| $T_5$ | 107 ms | 200 ms | 165 ms | 87 ms | M → N/O slow channel closing observed |
| $T_6$ | 4.4 s | 3.5 s | n.r. | 2.6 s | N/O → ACR1 return to ground state |

The determined half-lives $t_{1/2}$ obtained by FTIR and UV/VIS compared to electrophysiological[32] and UV/VIS measurements[24] from the literature. Half-lives $t_{1/2}$ were calculated from time constants $\tau$ for comparability. The last column presents the photocycle transitions and their corresponding physiological event. Intermediate assignments are based on spectroscopic measurements, and channel gating is assigned according to electrophysiological measurements. Note, the L-intermediate consists of three distinct states resulting in five possible kinetic models as described in Supplementary Fig. 5. Transitions that were not resolved by the applied technique are indicated by n.r.

processes within the photocycle; therefore, they are defined as marker bands. Marker bands were identified by global fit analysis of time-resolved FTIR absorbance data, which led to amplitude spectra (Supplementary Fig. 2, which reflect the spectral changes associated with the half-lives of the mathematically resolved transitions) and absorbance difference spectra (Supplementary Fig. 6, which reflect the changes at a given time point relative to the ground state). We identified marker bands that correspond to the $T_2$–$T_5$ transitions, representing the channel opening and closing; therefore, they are regarded as absorbance marker bands of channel gating. The band at $1691\,cm^{-1}$ predominantly raised at the same time with the fast channel opening and decayed upon channel closing. This band was assigned as a marker of the conducting states (Fig. 2a). The band at $1644\,cm^{-1}$ was assigned as a marker of the non-conducting states, as it corresponds to the decay and reformation of the non-conducting state. The $1644\,cm^{-1}$ band decayed in a two-step process, indicating both fast and slow channel opening. Both band positions were in a spectral region, suggesting structural changes in the amide I region (C=O backbone) of the protein, indicating global conformational changes over time. The band assignment was substantiated by our rapid-scan measurements of the D234N variant. A comparison of the D234N variant and WT in Supplementary Fig. 7a showed a significantly lower intensity of the difference band at $1644\,cm^{-1}$ in the variant, thus presenting fewer global conformational changes. Fewer conformational changes imply lower conductivity, which agrees with previous electrophysiological measurements showing significantly reduced conductivity for the D234N variant[26] compared with the WT. The absorbance changes at $1558\,cm^{-1}$ directly corresponded to the conducting state formation and decay (Supplementary Fig. 2).

**Marker bands for retinal conformation.** The $1184\,cm^{-1}$ band (Fig. 2a) is a protonated 13-*cis* retinal marker, analogous to bR[37]. Thus, the band decays upon the formation of the M intermediate state. Although its reformation in the N/O state was hardly visible, which indicates that reisomerization precedes reprotonation, it was observed in a detergent-solubilized sample (Supplementary Fig. 8). The measurement differences between the lipid and detergent environments are explained by the alteration of the N/O equilibria in different environments[38]. The $1529\,cm^{-1}$ band (Fig. 2a) reflects the C=C ground state retinal vibrations analogous to the bR ground state[39]. Both UV/VIS (Supplementary Fig. 4) and FTIR amplitude spectra (Supplementary Fig. 2) of M decay show the return of the ground state retinal bands. We interpret this as a reisomerization of the retinal in the O state, similar to the process observed in bR[40]. The N–O equilibrium seems to be mostly shifted toward the O state, leaving small amplitudes of the retinal bands for the N/O transition to the ground state.

**The mechanistic role of Glu-68.** We assigned the carbonyl band pair $1708(-)/1716(+)\,cm^{-1}$ (Fig. 2a and Supplementary Figs. 2, 3, 6, 9, and 10) to protonated Glu-68 based on our E68Q step-scan measurements. Supplementary Fig. 9 shows that the negative band at $1708\,cm^{-1}$ of the WT disappeared in E68Q. This band was assigned in accordance with previous low-temperature IR spectroscopic measurements at 170 K[34] as the band pair at $1708(-)/1724(+)\,cm^{-1}$ for a trapped L-intermediate in an L minus ground state spectrum. Based on the $1708\,cm^{-1}$ difference band, we conclude that Glu-68 is protonated in the ground state and deprotonated within sub-microseconds, early in the photocycle. Glu-68 was reprotonated in the $L_1$ to $L_2$ transition with a half-life of 18 µs alongside the conducting state formation. The correlation of Glu-68 reprotonation and pore formation was indicated by the concomitant decrease in $1708\,cm^{-1}$ band intensity and the emergence of the $1716\,cm^{-1}$ band within this 18 µs rate (Fig. 2 and Supplementary Figs. 3, 6, and 10). The hydrogen bond networks of the protonated Glu-68 changed between the ground state and the conducting state as $1708(-)/1716(+)\,cm^{-1}$ pair persisted throughout the conducting state. We therefore conclude that the $L_2$ state is trapped during the low-temperature measurements[34], which is in line with our data showing a band pair for protonated Glu-68. Furthermore, Glu-68 is deprotonated in $L_1$, as the positive band at $1716\,cm^{-1}$ was not observed in our measurements. The mechanistic role of Glu-68 is particularly interesting as it is conserved throughout all known ACRs[31], suggesting that the deprotonation/reprotonation mechanism associated with channel opening, proposed here, is a general principle applicable to all ACRs. However, additional FTIR studies on other ACRs are needed to verify this hypothesis.

**The mechanistic role of Asp-234.** Following low-temperature (77 K) IR and room temperature UV/VIS spectroscopic measurements, it was proposed that Asp-234 is protonated in the ground state[26]. However, the carbonyl band pair at $1740(-)/1732(+)\,cm^{-1}$, which was assigned to protonated Asp-234 in a previous study[26], was not present in our measurements at ambient temperature. In the carbonyl region, we observed two negative bands ($1708$ and $1725\,cm^{-1}$) that are candidates for the representation of protonated Asp-234. The $1708\,cm^{-1}$ band is assigned to protonated Glu-68, as previously mentioned. While the $1725\,cm^{-1}$ band was present in the D234N variant (Supplementary Fig. 7b), it shifted to $1732\,cm^{-1}$ and was less intense compared to that of the WT. In addition, the $1725\,cm^{-1}$ band was also shifted to $1732\,cm^{-1}$ in the E68Q variant (Supplementary Fig. 9). Therefore, the band could not be assigned to either protonated Glu-68 or Asp-234. Interestingly, the band at $1708\,cm^{-1}$ assigned to Glu-68 was not visible in D234N, indicating functional dependencies of Glu-68 and Asp-234. Due to the lack of a band that unequivocally represents protonated Asp-

234, we investigated the carboxylate region of the amplitude spectra (Supplementary Fig. 7a) for candidate bands to represent deprotonated Asp-234. Moreover, Supplementary Fig. 7c shows a positive band at 1401 cm$^{-1}$ in WT, which was not present in the difference spectra of the D234N variant. This band might represent deprotonated Asp-234 with a fluctuating hydrogen bond network during the photocycle. However, the negative counterpart of the band in the carboxylate region was not observed. The negative band might be weaker and not observable due to the low signal-to-noise ratio. Carboxylate bands are more challenging to identify than carbonyl bands, especially when observed due to an environmental change and not a protonation change. In addition to the current assumptions mentioned in the literature regarding protonated Asp-234, our measurements provide evidence for deprotonated Asp-234 in the ground state. To clarify this, we used molecular dynamics (MD) simulations to check the stability of potential Asp-234 protonation states in the two available ground state X-ray structures of GtACR1[26,27].

**Ground state structural network of Glu-68 and Asp-234.** To identify the ground state hydrogen bond networks of the central constriction site and the SB, we performed four 200 ns MD simulations of GtACR1 dimer structures embedded in a membrane. We combined our spectroscopic results on the protonation states with the two available ground state X-ray structures (protein data bank (PDB)-ID 6CSM[26] and 6EDQ[27]) to obtain four different starting structures for MD simulations. Based on the consistent experimental results, we protonated the SB and Glu-68. We checked both protonated and deprotonated Asp-234 since the spectroscopic results were not fully consistent.

The two MD simulations based on 6EDQ were equilibrated, whereas the ones of 6CSM were not, as demonstrated by the root mean square deviation analysis shown in Supplementary Fig. 11. Therefore, only the results based on 6EDQ were considered for further analysis.

Figure 2c shows the representative hydrogen bond network of the central constriction site obtained through MD simulations. The protonated Glu-68 forms a hydrogen bond with Asn-239. The hydrogen bond network of the central constriction site is stable independent of the Asp-234 protonation state (Supplementary Fig. 12).

Next, we checked the stability of the Asp-234 interaction network. Figure 2d shows that the hydrogen bond network between Asp-234, Tyr-72, and Tyr-207 was stable for monomers in our MD simulations of deprotonated Asp-234 but dissolved for protonated Asp-234 (Fig. 2e). The representative MD simulation structure of the Asp-234/Tyr-72/Tyr-207 interaction network corresponds to the X-ray structure within thermodynamic fluctuations for deprotonated Asp-234 and significantly varies for deprotonated Asp-234. These findings account for both monomers (Supplementary Fig. 13). The secondary structure analysis, shown in Supplementary Figs. 14 and 15, illustrates that the breaking of the Asp-234/Tyr-72 hydrogen bond and the corresponding conformational change of the Tyr-72 leads to distortion of the helix Tyr-72, whereas that helix remains stable for deprotonated Asp-234. Thus, the ground state X-ray structure 6EDQ analyzed through equilibrated MD simulations indicate, in agreement with our experimental data, a protonated SB and protonated Glu-68, but deprotonated Asp-234.

**GtACR1 does not enter a light-adapted state.** The light-adapted ground state P$_{480}$ of CrChR2 was best observed spectroscopically during the first few seconds following excitation after the intermediate states of the dark-adapted photocycle decayed. One characteristic feature of light-adapted P$_{480}$ is the C=N-syn

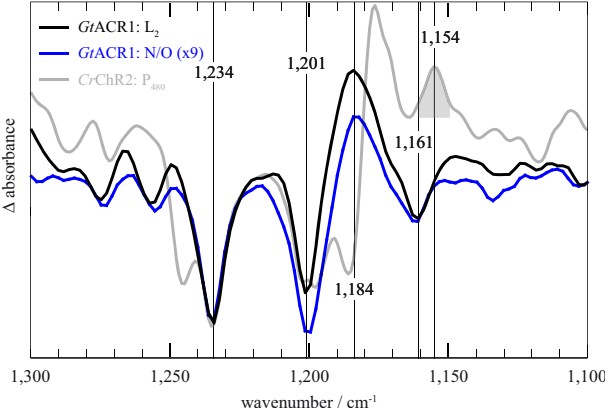

**Fig. 3 Differences in the retinal absorption bands between GtACR1 and CrChR2.** GtACR1 intermediate-minus-ground state difference spectra (L$_2$ in black, N/O in blue) compared to the P$_{480}$ difference spectrum of CrChR2 (gray) based on previously published data[15]. GtACR1 does not display a P$_{480}$ intermediate; thus, the CrChR2 P$_{480}$ spectrum is compared to the spectrum of a conducting state (L$_2$) and the late non-conducting N/O state. Spectra were scaled to the 1234 cm$^{-1}$ band. The N/O spectrum had to be amplified as indicated due to its low amplitude. The gray highlighted syn-band at 1154 cm$^{-1}$ of CrChR2 is not present in GtACR1, indicating that GtACR1 does not undergo a syn-cycle in contrast to CrChR2. The source data for this Fig. is given in Supplementary Data 1.

conformation of the retinal[15] represented by the conformation-specific marker band at 1154 cm$^{-1}$. However, GtACR1 displayed neither a late P$_{480}$-like intermediate nor the band at 1154 cm$^{-1}$ that has been assigned to the C=N-syn conformation in CrChR2 (Supplementary Fig. 16a and previous work[15]). Figure 3 shows that this band was absent from our measurement of GtACR1, which indicates that compared to CrChR2, the GtACR1 photocycle does not have a light-adapted state. To exclude the possibility that the C=N-syn conformation is represented at a different band position in GtACR1 compared to CrChR2, we performed FTIR-spectroscopic measurements with GtACR1 containing $^{13}C_{14}$–$^{13}C_{15}$ carbon-specific isotope-enriched retinal. The results (Supplementary Fig. 16b) indicated no C=N-syn conformation characteristic 14 cm$^{-1}$ downshift of the C$_{14}$–C$_{15}$ stretching vibration of the retinal[41,42], which was previously observed in CrChR2[15]. The expected isotope shifts are discussed in Supplementary Note 2. As further proof that the C=N-syn conformation band was not hidden or canceled out by overlapping difference bands, we recorded the Raman spectra of a photostationary state under continuous illumination (Supplementary Fig. 17b). The characteristic shifts of the retinal bands were observed; however, no C=N-syn species were identified under these conditions. The ground state spectra (Supplementary Fig. 17a) were highly similar to previously published ones[43]. For a detailed discussion of all isotope labeling experiments, refer to Supplementary Note 2.

Two key differences were observed in the photocurrent traces between GtACR1 and CrChR2. First, the former has a higher peak current and substantially lower inactivation. Structural data, accompanied by electrophysiological data[28], suggests that the preexisting tunnel within GtACR1, that is absent from CrChR2, evokes the higher peak current. Second, the syn-photocycle in CrChR2 contains the poorly conductive, long-lived open P$_{520}$ state, that leads to photocurrent inactivation in the CrChR2 photocycle. We conclude that the absence of a syn-photocycle in GtACR1, and the corresponding absence of a long-lived open state, causes low photocurrent inactivation. Meanwhile, the absence of a syn-photocycle in GtACR1 combined with its

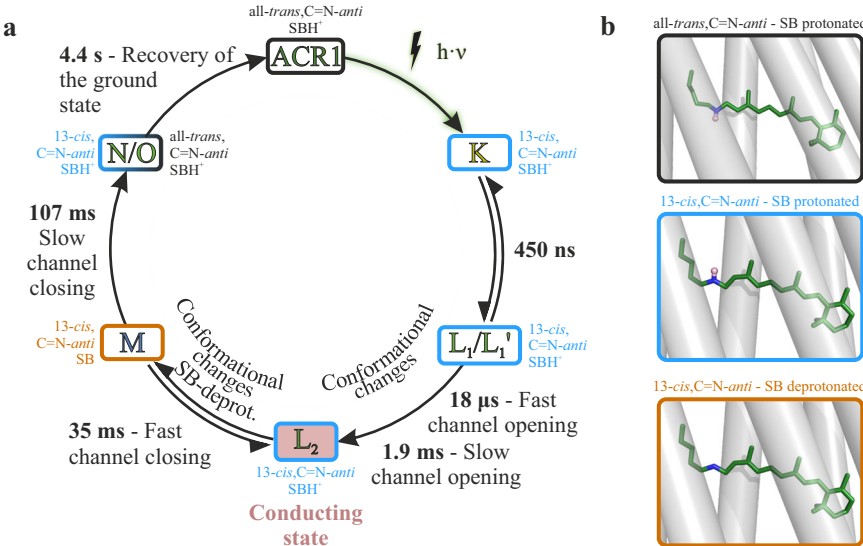

**Fig. 4 Revised photocycle model of *Gt*ACR1. a** The extended photocycle model is based on our ambient temperature time-resolved FTIR-spectroscopic data (Fig. 2) along with our UV/VIS spectroscopic data (Supplementary Fig. 4) and previously published electrophysiological measurements[32]. The K to L transition precedes channel opening indicating a non-conducting $L_1$ state. Channel opening occurs in a two-step process with a fast (18 μs) and a slow (1.9 ms) opening process to the conducting state $L_2$ (red). Therefore, $L_1$ was subdivided into $L_1$ and $L_1$'. The possible kinetic correlations of $L_1$ and $L_2$ are detailed in Supporting Fig. 5. Mechanistically, channel closing occurs exclusively upon M formation. However, the photocurrent fully disappears only upon M decay, resulting in a fast and a slow observed channel closing due to the reversible reaction between M and $L_2$. Next, *Gt*ACR1 slowly relaxes into the ground state from the N/O intermediate state. **b** Different retinal states during the photocycle. All-*trans*,C=N-*anti*-structure (black box) is taken from chain A of the X-ray structure of PDB-ID 6EDQ[27] and protonated according to our measurement. Structures in the blue and orange boxes are obtained by energy minimization of the retinal conformations with a restraint on the 13-*cis*,C=N-*anti*-conformation. These structures indicate the 13-*cis*,C=N-*anti*-retinal conformation with protonated and deprotonated Schiff base observed by our measurements. No C=N-*anti*-species is present.

preexisting tunnel explains the major differences in the photocurrents, namely the substantially reduced inactivation and the higher peak current of *Gt*ACR1 compared to *Cr*ChR2.

**Revised photocycle model**. The data above supports a revised *Gt*ACR1 photocycle model presented in Fig. 4. Compared to the previously published model[24], we provide a detailed description of the two-step conducting state formation by introducing a non-conducting $L_1/L_1$' state ahead of the conducting $L_2$ state, which was previously considered as the sole conducting L state[24]. Our revised model proposes that after photoexcitation, K is the first spectroscopically resolved intermediate state, exhibiting a 13-*cis*, C=N-*anti*-conformation of retinal. K decays to the non-conducting $L_1$ state within 450 ns. While the conducting $L_2$ state forms in two distinct kinetic steps, it is unclear whether $L_1$ and $L_1$' are in a mixed/inhomogeneous state or two consecutive states. In the first case $L_1$ and $L_1$' are two states, possibly in a slow equilibrium, co-arising from the K-intermediate and decaying to the same conducting $L_2$ state. If $L_1$ and $L_1$' are in equilibrium, we exclude a fast equilibrium between two closed states as there are two rates for channel opening. Meanwhile, in the second case, the transition from $L_1$ to $L_1$' forms a conducting pore, which becomes extended or altered following the $L_1$' → $L_2$ transition. Supplementary Fig. 5 describes five possible kinetic models and the transitions between the three distinct L states. A previous study suggested that the anion gate opening either occurs through two different mechanistic processes or two distinct gates, which open independently[32]. Nevertheless, independent of the anion gate opening, $L_2$ appears to be a homogeneous state as M formation and decay is described with single exponentials in FTIR and UV/VIS measurements, respectively. Channel closing occurs upon the deprotonation of the SB (M formation) and is observed as a two-step process due to the reversible reaction between $L_2$ and M, allowing $L_2$ to fully decay upon N/O formation. Channel closing

is accompanied by the decay of the negative band at 1644 cm⁻¹, indicating changes in the helices' backbone. The N/O equilibrium is far on the O-side, wherein the retinal is already reisomerized, which is analogous to bR, in which also the retinal reisomerizes during the N to O transition[40]. Finally, the N/O intermediate state decays back to the ground state with a half-life of 4.4 s.

## Conclusion

The channel opening process of the anion-conducting channelrhodopsin *Gt*ACR1 was monitored by time-resolved step-scan FTIR (Fig. 2 and Supplementary Figs. 2, 3, and 6) and UV/VIS (Supplementary Fig. 4) spectroscopy. The assignment of marker bands provides molecular insights into channel kinetics. The spectroscopically observed kinetic transitions (Table 1) are in agreement with previously published electrophysiologically recorded channel gating kinetics[32]. The integration of spectroscopic and electrophysiological data provides a detailed description of the channel gating step spectral transitions. This integrated perspective allowed us to gain deeper molecular insights (Fig. 4) into the photocycle of *Gt*ACR1. The L intermediates consist of an early non-conducting state $L_1$ and a late conducting state $L_2$. Marker bands of the protein backbone conformations for the non-conducting $L_1$ state and for conducting $L_2$ state were identified. However, no marker band was observed for C=N-*syn*. Therefore, a parallel *syn*-photocycle is ruled out for this anion-conducting channelrhodopsin. Nevertheless, the deprotonation of Glu-68 plays an important role in the *anti*-photocycle channel gating of *Gt*ACR1, despite the absence of the *syn*-photocycle. These molecular insights on *Gt*ACR1 provide an overall better understanding of the photocycle, suggesting strategies to improve cation ChRs for optimized optogenetic applications by constructing variants that lack the *syn*-cycle.

## Methods

### Biochemical methods

*Yeast culture.* *Pichia pastoris* strain of SMD1163 cells (kindly gifted by C. Bamann, Max Planck Institute of Biophysics, Frankfurt) containing the pPIC9k-ACR1-mCherry-His10 construct were precultured in buffered glycerol complex medium[44]. Expression of *Gt*ACR1 was induced using buffered methanol complex medium containing 2.5 μM all-*trans* retinal (either $^{12}$C or $^{13}$C$_{14}$,$^{13}$C$_{15}$-labeled) and 0.00004% biotin at an initial $OD_{600}$ of 1, at 30 °C and 120 rpm. Cells were harvested at an $OD_{600}$ of 20 by centrifugation.

*Membrane preparation and protein purification.* Cells were disrupted using a BeadBeater (Biospec Products), and membranes were isolated by ultracentrifugation. Homogenized membranes were solubilized overnight with 1% decylmaltoside. *Gt*ACR1 purification was performed by Ni-NTA affinity chromatography and removal of the mCherry fusion protein was achieved by TEV cleavage at 6 °C overnight. Subsequent gel filtration using a HiLoad 16/600 Superdex 200 pg column (GE Healthcare) yielded purified *Gt*ACR1.

*Reconstitution of GtACR1 into egg phosphatidylcholine.* Purified *Gt*ACR1 was reconstituted into egg phosphatidylcholine (Avanti Polar Lipids). The lipids were solubilized with 0.15% decylmaltoside in 20 mM HEPES (pH 7.5) and 100 mM NaCl by incubation at 50 °C for 10 min. Solubilized lipids and purified *Gt*ACR1 were mixed at a 2:1 ratio (lipid:protein, w-w) and incubated for 20 min. The detergent was removed overnight by adsorption on Bio-Beads SM 2 (BioRad) using 40:1 (Bio-Beads:detergent, w–w) at room temperature. The procedure of detergent removal was repeated the following day for 4 h. The resulting suspension containing proteoliposomes and buffer was then ultracentrifuged at 200,000×*g* for 2 h and the pellet was then transferred and squeezed between two $CaF_2$-slides to obtain an optical path length between 5 and 10 μm. This sample was then placed in a vacuum-tight cuvette.

### Spectroscopic methods

*FTIR-experiments.* To gain insight into the changes upon illumination, we performed time-resolved FTIR difference spectroscopy at 20 °C. The samples were illuminated with a short laser pulse of a Minilite Nd:YAG laser (Continuum, $\lambda_{max}$: 532 nm, 6 ns pulse). Interferograms were recorded using a Vertex 80 v spectrometer and OPUS 7.2 software (Bruker Corporation), an Adwin Pro II A/D converter and ADbasic 6 software (Jäger Computergesteuerte Messtechnik GmbH), and a Lecroy Waverunner HRO64zi oscilloscope with WaveRunner 6 Zi Oscilloscope Firmware version 6.6.0.5 (Teledyne LeCroy) and Matlab R2015a (The MathWorks, Inc.). Difference spectra were calculated using the Beer–Lambert law, which resulted in positive photo product bands and negative educt bands in the difference spectra. Step-scan spectra were obtained from a total of 111 measurements from eight protein samples, while rapid-scan spectra were obtained from approximately 750 measurements from one sample. Step-scan data and rapid-scan data were combined to generate the complete datasets (Supplementary Fig. 18).

*UV/VIS experiments.* UV/VIS samples were prepared using the same procedure as for the FTIR sample. Measurements were carried out using a monochromator and a photomultiplier. Samples were illuminated with a short laser pulse of a Minilite Nd:YAG laser (Continuum, $\lambda_{max}$: 532 nm, 6 ns pulse) and data were recorded using an Adwin Pro II (Jäger Computergesteuerte Messtechnik GmbH) system. UV/VIS spectra were obtained from approximately 100 measurements from one sample. Difference spectra were calculated using the Beer–Lambert law, which resulted in positive photo product bands and negative educt bands in the difference spectra.

*Raman spectroscopy.* Samples for Raman spectroscopy were prepared using the same procedure as for the FTIR samples. Raman spectroscopic analysis was performed using a WITec alpha300RA confocal Raman microscope with a single-frequency diode laser of 785 nm (Toptica Photonics AG) as the excitation source. A frequency-doubled Nd:YAG laser operating at a wavelength of 532 nm (Crystal Laser) was used for the excitation of *Gt*ACR1.

*Spectral data analysis.* IR data was analyzed using OPUS 7.2 (Bruker Corporation) and Matlab R2017b (The MathWorks, Inc.) software. UV/VIS data were analyzed using Matlab R2017b.

*Global fit analysis.* For the interpretation of the time-resolved data, it is assumed that the reaction cycles of the investigated proteins are composed of a sequence of first-order reactions. The error square *F* is a measure of the deviation of the recorded absorbances and the sum of the theoretical contributions from multi-exponential decays. The squared error function

$$F = \sum_{n=1}^{N_{\tilde{\nu}}} \sum_{m=1}^{N_t} w(\tilde{\nu}_n)^2 \left( \Delta E(\tilde{\nu}_n, t_m) - \left( a_\infty(\tilde{\nu}_n) + \sum_{i=1}^{N} a_i(\tilde{\nu}_n) e^{-k_i t_m} \right) \right)^2$$

is minimized in an iterative process, where $N_{\tilde{\nu}}$ is the number of wavenumbers in the spectra, $N_t$ is the number of time points, $w(\tilde{\nu}_n)$ are noise-dependent weighting factors, $\Delta E(\tilde{\nu}_n, t_m)$ is the measured absorbance difference for each wavenumber $\tilde{\nu}_n$

and time point $t_m$, N is the number of exponential functions, $k_i$ are the reaction rate constants, and the pre-factors of each exponential function $a_i(\tilde{\nu}_n)$ are the amplitude spectra and $a_\infty(\tilde{\nu}_n)$ are offsets.

**Computational methods.** MD simulations were performed using GROMACS 2019[45], and results were visualized using VMD 1.9.4[46] and PyMOL 2.2. Simulation trajectories were analyzed using the analysis tools of GROMACS 2019, VMD 1.9.4, and QwikMD[47]. Contact patterns were identified using PyContact[48] and the MAXIMOBY contact matrix algorithm.

*System setup.* The *Gt*ACR1 dimers from 6CSM (chains A and B)[26] and 6EDQ (chain A and B)[27] X-ray structures were prepared for simulation using the MOBY/MAXIMOBY program package (CHEOPS Molecular Modeling). The protein was protonated using Poisson–Boltzmann equation-based pK(a) calculation[49]. Next, the protein was inserted into a POPC membrane and oriented based on hydrophobicity calculations of a coarse-grained protein model using Lambada[50]. The protein–membrane interface was refined using g_membed[51]. Experimentally resolved protein internal water molecules were deleted, and water molecules of the first and second solvation shells of the protein–membrane system were added using the Vedani algorithm[52]. A cuboid simulation box with periodic boundary conditions filled with TIP4P bulk water molecules at a physiological salt concentration of 154 mmol/L, using sodium cations and chlorine anions was generated with GROMACS[45]. The ion ratio was adjusted to ensure the neutral charge of the simulation system. MAXIMOBY was used to identify energetically unfavorable side-chain and backbone conformations as well as bulk water molecules, which were subsequently locally optimized using the conjugate gradient algorithm with the implemented Amber99 force field.

Classical molecular mechanic simulations were performed with GROMACS 2019 using the optimized potential for liquid simulations all-atom (OPLSA/AA) force field. For the POPC bilayer, previously reported parameters were used[53]. Electronic interactions were determined by the Fast Particle-Mesh Ewald method using a grid spacing of 0.12 nm, fourth-order interpolation, and a Coulomb cut-off value of 1.0 nm. For van der Waals interactions, a cut-off value of 1.0 nm was used. Covalent hydrogen bonds were constrained to their equilibrium length using the LINCS algorithm[54].

*Simulation run parameters.* The system was heated up over a course of 100 ps in 1 fs time steps with a linear gradient from 0 to 100 K and then further heated up within 1 ns to 310 K (body temperature). Next, a 500 ps (2 fs time steps) NVT equilibration run (number of atoms *n*, volume *V*, and temperature *T* are constant and the pressure is variable) was performed followed by a 25 ns NPT equilibration run (number of atoms *n*, pressure *p*, and temperature *T* are constant and the volume is variable). Pressure and temperature were kept at constant values of 1 standard atmosphere and 310 K, respectively, using a Berendsen barostat with a coupling constant $\tau_p$ of 0.5 ps and a modified Berendsen thermostat (V-rescale) with a coupling constant $\tau_T$ of 0.1 ps. NEXT, a 100 ns (2 fs time step) production run was performed using Parinello–Rahman with a $\tau_p$ of 2.5 ps for pressure control and Nose–Hoover with a $\tau_T$ of 0.5 ps for temperature control.

**Statistics and reproducibility.** Step-scan spectra were obtained from a total of 111 measurements from 8 samples, rapid-scan spectra from about 750 measurements from one sample, UV/VIS spectra from about 100 measurements from one sample. Raman spectra were obtained by averaging 2500 individual spectra from one sample. The number of repetitions was adjusted according to data quality. IR-Measurements with large baseline drifts of the interferograms were excluded.

**Reporting summary.** Further information on research design is available in the Nature Research Reporting Summary linked to this article.

## Data availability

The raw data of the spectroscopic measurements as well as the simulation trajectories and input files will be provided upon individual request by the corresponding authors. The source data for the main figures is given in Supplementary Data 1.

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

## Acknowledgements
We thank Elena Govorunova for providing the raw electrophysiological data for Supplementary Fig. 1. We thank Udo Höweler for fruitful discussion of the structural results. We acknowledge Harald Chorongiewski for his technical support with the spectroscopic setup and Gabriele Smuda for molecular biology support. This work was supported by Deutsche Forschungsgemeinschaft (DFG, German Research Foundation) Individual Research Grant, "Molecular mechanisms of cation and anion-conducting channelrhodopsins" (GE 599/23-1) to K.G. and the DFG Priority Program SPP1926 (GE 599/19-2 and GE 599/19-1) to K.G. Further support was provided by the Ministry for Culture and Science (MKW) of North Rhine-Westphalia (Germany) through grant 111.08.03.05-133974 to K.G. and the Protein Research Unit Ruhr within Europe (PURE) funded by the Ministry of Innovation, Science and Research (MIWF) of North-Rhine Westphalia (Germany) to K.G.

## Author contributions
K.G. obtained the funding; T.R. and K.G. designed the research; M.-A.D., P.A., M.J.N., S.A.T., S.F.E.-M., and T.R. performed the research; M.-A.D. performed FTIR and UV/VIS measurements with the help of M.J.N. supervised by C.K., T.R., and K.G.; M.-A.D. performed biochemistry supervised by M.L.; P.A. and S.A.T. performed the MD simulations supervised by T.R.; M.-A.D. and S.F.E.-M. performed Raman measurements; all authors analyzed the data; M.-A.D., M.L., C.K., T.R., and K.G. wrote the paper with edits from all co-authors.

## Funding

## Competing interests

The authors declare no competing interests.
