## [Peer Review File · Communications Biology]

Reviewer #1:

Remarks to the Author:

CRITIQUE

The authors present an expert, thorough, and incisive analysis primarily by time-resolved step-scan FTIR and UV-vis flash photolysis of the photochemical reaction cycle of the light-gated anion channelrhodopsin *GtACR1*. The *GtACR1* protein is of great interest as the most conductive channelrhodopsin under study and also its use as one of the most effective optogenetic tools.

The authors report notable advances in our understanding of channelrhodopsin mechanisms. Their results provide valuable new information regarding the photocycle transitions in *GtACR1*, and correlate with greater time-resolution the relationship of channel opening and closing to photocycle chemical events. They demonstrate a major difference not previously known between the photoreactions of the most studied channelrhodopsin, *CrChR2*, and those of *GtACR1*, namely the absence of a C=N-*syn* species in the latter. In *CrChR2* a C=N-*syn* subpopulation photocycling correlates with a poorly conductive state and appears to be a significant factor in inactivation of the protein in optogenetics protocols.

Innovative results are the identification of marker absorption bands in the infrared spectral region. The FTIR bands provide new parameters for analysis of kinetics of crucial processes, including Schiff base proton transfers and helix movements that accompany channel opening and closing. The methodology developed by the authors has the advantage of being "all optical" thereby enabling monitoring kinetics of photocycle transitions and channel functions in the same molecules in the same conditions.

In summary, the findings significantly advance the study of *GtACR1* and channelrhodopsins in general. The manuscript is closely reasoned and well written. I recommend the submission with high enthusiasm and have only one suggestion for improvement:

1) Page 5, Lines 101 through 107: "Despite their overall structural similarity, *CrChR2* and *GtACR1* exhibit several marked differences. ..." Differences from the literature are discussed here and elsewhere and include the authors' interesting discovery of a previously unknown difference, namely the lack in *GtACR1* of a C=N-*syn* species, which the authors suggest may account for the greater inactivation rate in *CrChR2*. *GtACR1* has 20-fold greater unitary conductance and far less inactivation as *CrChR2*. Absence of a C=N-*syn* species may improve its efficiency, and that is reasonable to suggest. However, another difference likely to contribute to the more robust and persistent photocurrents from *GtACR1* is its preexisting tunnel evident in the dark crystal structure that traverses the molecule from extracellular to cytoplasmic pores of the molecule (Authors' reference 23). No such tunnel is evident in *CrChR2* (Authors' reference 21) nor any other channelrhodopsins with known atomic structures. The preexisting tunnel is largely open in the dark with 3 distinct constrictions that block anion flux, suggesting a larger channel in *GtACR1* that may account for greater conductance and its persistence. For comparison of the 2 channelrhodopsins, absence of a C=N-*syn* species is interesting, but the presence in *GtACR1* of a preexisting tunnel should also be added to the discussion.

Reviewer #2:

Remarks to the Author:

Cation and anion channelrhodopsins have been in the focus of intensive research especially because of their potential for optogenetic applications. Archetypical channel rhodopsins from *Chlamydomonas reinhardtii* are cation channels. A mutation (E90Q) convert CrChR2 into a Cl⁻ channel. Recently, naturally occurring anion channels with very interesting optogenetic properties were identified in various organisms. These were mainly analysed J. Spudich and colleagues. Using a thorough biophysical approach, Dreier et al. describe kinetic data from GtACR1 derived from UV/Vis and FTIR experiments. These experimental results are substantiated by MD calculations. The main conclusions of this work concern two major points. Contrary to CrChR2, GtACR1 displays solely an anti photocycle, which might be a reason for a more efficient channel behaviour. A second important conclusion concerns the kinetic of channel opening which the authors place at the L2 state. In accordance with work from the Spudich lab, L2 is formed in two distinct kinetic steps (18 μ s, 1.9 ms). Glu-68 (Glu-90 in CrCh2) being protonated in the groundstate deprotonates in the early steps of the photocycle but is reprotonated with 18 μ s. These two kinetic steps are correlated with the fast and slow channel opening. The fast and slow channel closing events are correlated with L2 \rightarrow M and M \rightarrow N/O transitions. These results are certainly of interest. However, there are some points, which need clarification before the paper can recommended for publication.

- The state of L has to be explained in more detail: In table 1, T2 and T3 are defined as 'within L' whereas the model denotes a L1/L'1 state. These two designations are somehow confusing. Are the two L1 states in a fast equilibrium? How can one envision a kinetic model in which L2 is formed with two time constants? What about the relative amplitudes? According to Sineshchekov et al. (ref. 29) the mutant E68Q influences the kinetics of channel opening and closing. Photocycle data derived from this mutant would be important.

- A comparison of newly described sequences might provide further insights into the molecular mechanism of channel opening and closing. All natural occurring ACRs contain at this position still a Glu residue (Glu-68 in GtACR1). This should be discussed in line with the results presented here.

- The paragraph headed conclusion rather corresponds to a summary of the previous section. The paper would gain if the authors would discuss their data in relation to current literature on ACRs. E.g. E. Govorunova et al. (2018) described fast and slow variants of cryptophyte anion channelrhodopsins. Interestingly, Cys residues in helix 3 (Cys-95 in GtACR1) influences the channel kinetics. Are their indications to be found in the FTIR data?

- Citation 4: the original discovery of ChRs should be quoted.

- First paragraph: Optogenetic tools not only comprise microbial rhodopsins but also other protein families (e.g. phytochromes).

- 2nd paragraph: prototype \rightarrow better archetype

Reviewer #1

1.a: *“In summary, the findings significantly advance the study of GtACR1 and channelrhodopsins in general. The manuscript is closely reasoned and well written. I recommend the submission with high enthusiasm and have only one suggestion for improvement:*

1) Page 5, Lines 101 through 107: “Despite their overall structural similarity, CrChR2 and GtACR1 exhibit several marked differences. ...” Differences from the literature are discussed here and elsewhere and include the authors’ interesting discovery of a previously unknown difference, namely the lack in GtACR1 of a C=N-syn species, which the authors suggest may account for the greater inactivation rate in CrChR2. GtACR1 has 20-fold greater unitary conductance and far less inactivation as CrChR2. Absence of a C=N-syn species may improve its efficiency, and that is reasonable to suggest. However, another difference likely to contribute to the more robust and persistent photocurrents from GtACR1 is its preexisting tunnel evident in the dark crystal structure that traverses the molecule from extracellular to cytoplasmic pores of the molecule (Authors’ reference 23). No such tunnel is evident in CrChR2 (Authors’ reference 21) nor any other channelrhodopsins with known atomic structures. The preexisting tunnel is largely open in the dark with 3 distinct constrictions that block anion flux, suggesting a larger channel in GtACR1 that may account for greater conductance and its persistence. For comparison of the 2 channelrhodopsins, absence of a C=N-syn species is interesting, but the presence in GtACR1 of a preexisting tunnel should also be added to the discussion.”

We appreciate your encouraging feedback on our manuscript. We agree that the discussion of the preexisting tunnel can be further elaborated. Accordingly, we have included a statement regarding the

preexisting tunnel structural differences between GtACR1 and CrChR2 within the Introduction, as follows (page 5, line 102-109):

“Despite their overall structural similarity, CrChR2²⁵ and GtACR1^{26,27} exhibit several marked differences. For instance, **GtACR1 contains a preexisting tunnel²⁶⁻²⁸ that is not present in CrChR2²⁵**. In addition to a low sequence identity (24%), the crucial residues in CrChR2 that are functionally responsible for the inner gate (Glu-82, Glu-83, His-134, and Arg-268)²⁵, the SB counter ion (Glu-123)^{29,30}, and a prominent key residue for the syn-cycle (Lys-93)¹⁵, are absent from GtACR1. In contrast, a critical player in the CrChR2 photocycle, Glu-90¹⁵⁻¹⁷, is conserved in GtACR1 (Glu-68) and other known ACRs³¹, although absent in bR.”

The preexisting tunnel actually perfectly complements our results as it explains the substantially higher peak current in GtACR1 that cannot be explained by the absence of the *syn*-photocycle. Taking these two findings together they provide explanations for the two key differences in the photocurrent, namely the higher peak current and lower inactivation of GtACR1 compared to CrChR2. According to your suggestion, we have included the following description in the revised manuscript, as follows (page 15, line 337-346):

“Two key differences were observed in the photo current traces between *GtACR1* and *CrChR2*. First, the former has a higher peak current and substantially lower inactivation. Structural data, accompanied by electrophysiological data²⁸, suggests that the preexisting tunnel within *GtACR1*, that is absent from *CrChR2*, evokes the higher peak current. Second, the *syn*-photocycle in *CrChR2* contains the poorly conductive, long-lived open P₅₂₀ state, that leads to photocurrent inactivation in the *CrChR2* photocycle. We conclude that the absence of a *syn*-photocycle in *GtACR1*, and the corresponding absence of a long-lived open state, causes low photocurrent inactivation. Meanwhile, the absence of a *syn*-photocycle in *GtACR1* combined with its preexisting tunnel explains the major differences in the photocurrents, namely the substantially reduced inactivation and the higher peak current of *GtACR1* compared to *CrChR2*.”

Reviewer #2

2.a: *The state of L has to be explained in more detail: In table 1, T2 and T3 are defined as ‘within L’ whereas the model denotes a L1/L’1 state. These two designations are somehow confusing.*

We agree with this comment. There are at least five possible kinetic models that describe the T2 and T3 transitions within three distinct L states. To clarify this in Table 1 we have added Supplementary Figure 5 describing the kinetic models, which has been referenced in Table 1 caption.

2.b: *Are the two L1 states in a fast equilibrium? How can one envision a kinetic model in which L2 is formed with two time constants?*

It is unclear whether L₁ and L₁’ are a mixed/inhomogeneous state or two consecutive states. Two closed states L₁ and L₁’ cannot be in a fast equilibrium as that would have only one rate for channel opening, however, two were observed. For clarification we have also added Supplementary Figure 5 as well as the following description to the revised manuscript. (page 16, lines 372-379):

“While the conducting L₂ state forms in two distinct kinetic steps, it is unclear whether L₁ and L₁’ are in a mixed/inhomogeneous state or two consecutive states. In the first case L₁ and L₁’ are two states, possibly in a slow equilibrium, co-arising from the K-intermediate and decaying to the same conducting L₂ state. If L₁ and L₁’ are in equilibrium, we exclude a fast equilibrium between two closed states as there are two rates for channel opening. Meanwhile, in the second case, the transition from L₁ to L₁’ forms a conducting pore, which becomes extended or altered following the L₁’ → L₂ transition.

Supplementary Figure 5 describes five possible kinetic models and the transitions between the three distinct L states.”

2.c: *What about the relative amplitudes?*

The amplitudes of the marker bands for channel opening (Fig. S2) are larger for the faster rate, which agrees with the electrophysiological data. This has been clarified in the revised manuscript as follows (page 7, lines 157-160)

“The amplitudes of the marker bands representing channel opening (T_2 and T_3) are larger in the faster rate, which agrees with the electrophysiological data that showed the first rate of channel opening to be more prominent than the second rate³⁵.”

2.d: *According to Sineshchekov et al. (ref. 29) the mutant E68Q influences the kinetics of channel opening and closing. Photocycle data derived from this mutant would be important.*

We agree that the E68Q mutant represents a crucial mutant and are currently working on time-resolved data for this mutant to further elucidate the events taking place during the opening and closing of the GtACR1 channel. However, this is beyond the scope of the current manuscript and will instead be the subject of follow-up studies.

2.e: *A comparison of newly described sequences might provide further insights into the molecular mechanism of channel opening and closing. All natural occurring ACRs contain at this position still a Glu residue (Glu-68 in GtACR1). This should be discussed in line with the results presented here.*

We completely agree with this comment. Accordingly, we have further strengthened this point in the revised manuscript, as follows (page 12, lines 248-252):

“The mechanistic role of Glu-68 is particularly interesting as it is conserved throughout all known ACRs³¹, suggesting that the deprotonation/reprotonation mechanism associated with channel opening, proposed here, is a general principle applicable to all ACRs. However, additional FTIR studies on other ACRs are needed to verify this hypothesis.”

We have also included the following citation in the introduction on page 5, lines 107-109:

“In contrast, a critical player in the CrChR2 photocycle, Glu-90¹⁵⁻¹⁷, is conserved in GtACR1 (Glu-68) and other known ACRs³¹, although absent in bR.”

2.f: *The paragraph headed conclusion rather corresponds to a summary of the previous section. The paper would gain if the authors would discuss their data in relation to current literature on ACRs. E.g. E. Govorunova et al. (2018) described fast and slow variants of cryptophyte anion channelrhodopsins. Interestingly, Cys residues in helix 3 (Cys-95 in GtACR1) influences the channel kinetics. Are their indications to be found in the FTIR data?*

The impact of helix three on channel kinetics is indeed important. As there is no Cys-95 in GtACR1 we assume that you were referring to Cys-102, which is in fact an interesting residue. We measured the absorption region from 900 to 1900 cm^{-1} and this region does not include the absorption band of cysteine ($\sim 2550 \text{ cm}^{-1}$). Hence, this data does not provide any additional useful information regarding crucial cysteine residues. To study the impact of cysteine residues the spectroscopic setup must be modified to broaden the measured absorption region. Although such modifications are beyond the

scope of the current manuscript, they will be a very interesting topic to follow up with in future studies. Nevertheless, according to your suggestion, we have broadened the discussion of the literature (including *E. Govorunova et al. (2018)*) and added the role of Cys-102 to the discussion of the channel kinetic in the revised Introduction on page 5 lines 109-111:

“Moreover, within GtACR1, Glu-68 together with Asp-234 and Cys-102, may participate in channel gating as their mutations influence channel kinetics^{24,32}; interestingly C102A evokes bistability of GtACR1³³.”

24. Sineshchekov, O. A., Li, H., Govorunova, E. G. & Spudich, J. L. Photochemical reaction cycle transitions during anion channelrhodopsin gating. *Proc. Natl. Acad. Sci. U.S.A.* **113**, E1993-2000 (2016).
32. Sineshchekov, O. A., Govorunova, E. G., Li, H. & Spudich, J. L. Gating mechanisms of a natural anion channelrhodopsin. *Proc. Natl. Acad. Sci. U.S.A.* **112**, 14236–14241 (2015).
33. Govorunova, E. G. *et al.* Extending the Time Domain of Neuronal Silencing with Cryptophyte Anion Channelrhodopsins. *eNeuro* **5**, (2018).

- Citation 4: the original discovery of ChRs should be quoted.

We have modified the sentence to account for the discovery of channelrhodopsins as light-gated ion channels (page 3, lines 37-39) and cited the following literature:

“The initial discovery of channelrhodopsins (ChRs) in *Chlamydomonas reinhardtii*, as well as their identification as light-activated ion channels, represent the inception of optogenetics⁴⁻⁹.”

4. Fuhrmann, M., Stahlberg, A., Govorunova, E., Rank, S. & Hegemann, P. The abundant retinal protein of the *Chlamydomonas* eye is not the photoreceptor for phototaxis and photophobic responses. *J. Cell Sci.* **114**, 3857–3863 (2001).
5. Sineshchekov, O. A., Jung, K.-H. & Spudich, J. L. Two rhodopsins mediate phototaxis to low- and high-intensity light in *Chlamydomonas reinhardtii*. *Proc. Natl. Acad. Sci. U.S.A.* **99**, 8689–8694 (2002).
6. Nagel, G. *et al.* Channelrhodopsin-1: A Light-Gated Proton Channel in Green Algae. *Science* **296**, 2395–2398 (2002).
7. Nagel, G. *et al.* Channelrhodopsin-2, a directly light-gated cation-selective membrane channel. *Proc. Natl. Acad. Sci. U.S.A.* **100**, 13940–13945 (2003).
8. Suzuki, T. *et al.* Archaeal-type rhodopsins in *Chlamydomonas*: model structure and intracellular localization. *Biochem. Biophys. Res. Commun.* **301**, 711–717 (2003).
9. Bregestovski, P. & Mukhtarov, M. Optogenetics: Perspectives in Biomedical Research (Review). *Sovrem. Tehnol. Med.* **8**, 212–221 (2016).

2.g: *First paragraph: Optogenetic tools not only comprise microbial rhodopsins but also other protein families (e.g. phytochromes).*

Indeed. The following was added in the first paragraph on page 3, lines 41-44:

“In addition to the light-activatable channels, various ion pumps comprising the microbial rhodopsin superfamily¹, as well as caged effector proteins that use LOV domains, or dimerizing systems that

employ phytochromes or cryptochromes to stimulate protein-protein interactions¹¹, are utilized as optogenetic tools.”

11. Repina, N. A., Rosenbloom, A., Mukherjee, A., Schaffer, D. V. & Kane, R. S. At Light Speed: Advances in Optogenetic Systems for Regulating Cell Signaling and Behavior. *Annu. Rev. Chem. Biomol. Eng.* **8**, 13–39 (2017).

2.h: *2nd paragraph: prototype → better archetype*

We have revised the text accordingly (page 3, line 53).

REVIEWERS' COMMENTS:

Reviewer #1 (Remarks to the Author):

Changes made by the authors improved this fine manuscript. I have no further changes other than the following two minor points:

1) Lines 103-104: "...GtACR1 contains a preexisting tunnel^{26–28} that is not present in CrChR..."

Reference 26 (Kim et al 2018) should not appear in this sentence, since Kim et al did not report or mention the tunnel, which was first revealed by analytical techniques applied by reference 27 (Li et al 2019). It would be correct to cite references 27 and 28 after "tunnel".

2) In several appearances of "GtACR1" , the "Gt" needs to be italicized.

Congratulations to the authors for their excellent contribution.

Reviewer #2 (Remarks to the Author):

The authors have convincingly answered the points I raised and have altered the manuscript accordingly. I can recommend the paper for publication.